# Levamisole in Children with Idiopathic Nephrotic Syndrome: Clinical Efficacy and Pathophysiological Aspects

**DOI:** 10.3390/jcm8060860

**Published:** 2019-06-16

**Authors:** Anne K. Mühlig, Jun Young Lee, Markus J. Kemper, Andreas Kronbichler, Jae Won Yang, Jiwon M. Lee, Jae Il Shin, Jun Oh

**Affiliations:** 1Department of Pediatrics, University Hamburg-Eppendorf, 20246 Hamburg, Germany; a.dettmar@uke.de (A.K.M.); j.oh@uke.de (J.O.); 2Department of Nephrology, Yonsei University Wonju College of Medicine, Wonju, Kangwon 26426, Korea; junyoung07@yonsei.ac.kr (J.Y.L.); kidney74@yonsei.ac.kr (J.W.Y.); 3Department of Pediatrics, Asklepios Klink Nord-Heidberg, 22417 Hamburg, Germany; m.kemper@asklepios.com; 4Department of Internal Medicine IV (Nephrology and Hypertension), Medical University Innsbruck, Innsbruck 6020, Austria; andreas.kronbichler@i-med.ac.at; 5Department of Pediatric Nephrology, Chungnam National University Hospital, Daejeon 35015, Korea; jwmleemd@gmail.com; 6Department of Pediatrics, Yonsei University College of Medicine, Seoul 03722, Korea; 7Division of Pediatric Nephrology, Severance Children’s Hospital, Seoul 03722, Korea; 8Institute of Kidney Disease Research, Yonsei University College of Medicine, Seoul 03722, Korea

**Keywords:** levamisole, nephrotic syndrome, podocyte, steroid-dependent nephrotic syndrome

## Abstract

Steroid sensitive nephrotic syndrome is one of the most common pediatric glomerular diseases. Unfortunately, it follows a relapsing and remitting course in the majority of cases, with 50% of all cases relapsing once or even more often. Most children with idiopathic nephrotic syndrome respond initially to steroid therapy, nevertheless repeated courses for patients with relapses induce significant steroid toxicity. Patients with frequent relapses or steroid dependency thus require alternative treatment, such as cyclophosphamide, cyclosporine, tacrolimus, mycophenolate mofetil, levamisole, or rituximab. To reduce the relapse rate, several drugs have been used. Among these, levamisole has been considered the least toxic and least expensive therapy. Several randomized controlled trials (RCT) showed that levamisole is effective in reducing the relapse risk in steroid sensitive forms of nephrotic syndrome with a low frequency of side effects. Levamisole is a synthetic imidazothiazole derivative with immune-modulatory properties. In this article, we review recent data from randomized trials and observational studies to assess the efficacy of levamisole in frequently relapsing nephrotic syndrome and steroid-dependent nephrotic syndrome.

## 1. Introduction

Nephrotic syndrome (NS) occurs in 16 out of 100,000 children in western countries and is one of the most common renal diseases in the pediatric population [1]. It is characterized by the appearance of proteinuria (>1 g/m^2^/day), hypoalbuminemia (<25 g/L), generalized edema and normal glomerular function. Usually, there is an idiopathic NS (iNS) with two histological subtypes: (1) NS with small histological changes, so-called minimal change nephrotic syndrome (MCNS); and with more pronounce histological lesions, (2) focal segmental glomerulosclerosis (FSGS).

Idiopathic nephrotic syndrome (iNS) seems to be a T-cell-mediated disease. Clinical observations show that the first episode and relapses often develop after an infection of the upper respiratory tract or an allergic reaction. In the acute phase of the disease, a shift in lymphocyte population has been reported [2]. There also seems to be an enhanced type II immunologic reaction in inactive forms of NS [3]. These findings and others indicate that there has to be an immunologic trigger in patients with iNS.

Centrally involved in the onset of proteinuria are podocytes [4]. Podocytes are highly specialized, terminally differentiated cells, forming the glomerular filtration barrier together with the glomerular basement membrane and glomerular endothelial cells. Effacement of podocyte foot processes can be seen in electron microscopy, which is the only abnormality seen in MCNS.

Treatment with steroid is still golden standard, and most forms (90%) of NS show remission within four weeks of treatment with steroids. In cases who respond to steroids, iNS is classified as steroid-sensitive nephrotic syndrome (SSNS), defined by the International Study of Kidney Disease in Children (ISKDC) [5].

The overall prognosis of children suffering from NS is good. Usually, the disease generally disappears at the onset of puberty. Nevertheless, up to 85% develop at least one relapse within the next five years [6]. Relapse risk generally decreases over time, but is prolonged during adulthood in some cases [7]. Recent research focuses on factors leading to initiation of iNS, the prediction of recurrence rate and important factors to reduce the risk of relapse. However, the exact mechanism of iNS are still elusive.

## 2. Side Effects of Traditional Immunosuppressive Agents in the Treatment of Nephrotic Syndrome

Regarding the literature, approximately 30–60% of all children suffering from a SSNS are frequently relapsing or even steroid-dependent forms of NS [8,9]. Frequently relapsing nephrotic syndrome (FRNS) is defined as more than four relapses in one year or more than two relapses following the first six months after the initial presentation. Patients with at least two relapses during treatment with alternate-day steroids or within 14 days after stopping steroid treatment are classified as steroid-dependent (SDNS) [8,10]. Treatment of relapses is usually based on steroids with a shorter duration (prednisone 60 mg/m^2^ until remission, 4 weeks 40 mg/m^2^ every other day) [5]. FRNS and SDNS usually react quickly to treatment with glucocorticoids, and long-term prognosis regarding renal function is very good [11], but patients suffer from relapses frequently and need to be treated repeatedly. However, cushingoid changes, hyperglycemia, infection, abnormal bone metabolism, skin atrophy, striae, peptic ulcer, adrenal suppression, increased blood pressure, and behavioral changes are well-recognized, potentially serious adverse effects of corticosteroids [12,13,14].

Therefore, there is an increasing demand for alternative less harmful therapies. In the past, there has been extensive evidence for the use of so-called steroid-sparing agents such as mycophenolic acid (MPA), rituximab, alkylating agents like cyclophosphamide and calcineurin inhibitors, which have been summarized and reviewed elsewhere [5,15]. Cyclophosphamide has bladder toxicity and other side effects such as gonadal toxicity, bone marrow depression, carcinogenesis, and an increased risk of infection [16]. Calcineurin inhibitors, cyclosporine and tacrolimus may lead to neurotoxicity, hirsutism, gingival hyperplasia, and nephrotoxicity [17]. Mycophenolate mofetil (MMF) and MPA are newer immunosuppressive drugs that have fewer serious adverse effect but have the disadvantage of high therapy costs [18,19].

The aim of this review is to focus on the use of levamisole, which seems to have a similar effect with other substances, but exhibits a lower number of adverse events and reactions. Levamisole has been considered the least toxic and least expensive steroid-sparing drug for preventing relapses of SSNS. Therefore, the use of levamisole should always be taken into account. However, levamisole is used in restricted areas (mostly in India and Europe). In FRNS and SDNS, levamisole significantly reduces both the relapse rate and the cumulative steroid dose; therefore, it should be recommended for these patients. In this review, we focus on the efficacy by critically discussing the literature, including randomized controlled trials (RCT), observational studies and existing meta-analyses for levamisole in order to highlight the usefulness and side effects of the drug, and the implications to treatment for nephrotic syndrome.

## 3. The Use of Levamisole in the Treatment of Nephrotic Syndrome

Levamisole is an immune-modulating imidazothiol-derived anthelminthic. It was first used in 1969 as an anthelmintic agent. In humans, levamisole was originally used in the treatment of leprosy [20] and colonic carcinoma [21]. As a treatment of SSNS, it has been used since the early 1980s [22]. Besides a few uncontrolled levamisole studies [23,24], the first controlled study was performed by the British Association of Pediatric Nephrology, published in 1991. In this study, a significant reduction of disease relapses in children was demonstrated with the use of levamisole [25]. Data from Kemper et al. identified that patients with exclusively frequent relapses responded to levamisole better than those with steroid dependence [26]. The Indian pediatric nephrology group revised its guidelines of nephrotic syndrome so that levamisole could be used as a steroid-sparing agent. The recommended dose is 2–2.5 mg/kg on alternate days for 12–24 months, while monitoring leukocyte count every 12–16 weeks [27]. In 2017, Gruppen et al. published an international RCT. In this double-blinded study, 99 children with SDNS and FRNS were treated with levamisole (2.5 mg/kg on alternate days) or placebo. They found that there was a significantly longer time to the first relapse in patients treated with levamisole compared to the placebo group [28]. Subgroup analysis showed that cases with SDNS (mostly from Europe) had an inferior response to FRNS (mostly from India). There are some considerations preferring levamisole in the treatment of FRNS over its use in SDNS [20,29]. The authors discussed different genetic backgrounds and immunological changes due to steroid dependency and preceding long-term steroid treatments, resulting in pharmacological interactions [28,29]. Recently in an open-label RCT studying SDNS and FRNS, both MMF (76 patients with 750–1000 mg/m^2^) and levamisole (73 patients with 2 to 2.5 mg/kg on alternate days) showed similar relapse rates, treatment failure rates, disease course and decreased cumulative steroid dose [30]. As 2.5 mg/kg every other day is the historical dosage, more studies are needed to define the appropriate dosage and duration of treatment [29].

There are several older RCTs, which were summarized in a Cochrane analysis published in 2013 [31]. When treatment with levamisole was compared to placebo, low-dose prednisone or no treatment, there was a significant reduction of relapse numbers in children with SSNS. Several different levamisole dosages have been employed in terms of total dose administrated (35 mg/m^2^/month vs. 20 mg/m^2^/month) or frequency (alternate day vs. 2 consecutive days of 7 days) [32]. The risk of relapse during the treatment period decreased by up to 10–45% in various studies [33]. However, the duration of treatment was relatively short; it varied from four to twelve months. Another observation was that levamisole reduced the relapse rate only during the actual treatment period, and had no significant impact on relapse risk after the discontinuation. Compared to intravenous, monthly [34] or orally administered cyclophosphamide [31], there seems to be a slightly higher rate of relapse during levamisole therapy. Nevertheless, there was no significant difference in the number of children suffering from one or more relapses in the period between both treatments [24].

In addition to these RCTs, there have been a lot of smaller retrospective studies in the last decade, but most often only single center experiences have been published [35,36,37,38,39,40]. These papers stated that in children with FRNS or SDNS [36,37,38,40], levamisole could be effective in decreasing relapse risk and frequency. However, some of these studies used low-dose prednisolone or a slowly tapering regimen in addition to 2–2.5 mg/kg body weight levamisole on alternate days [27]. Many of these studies showed that levamisole lowered the number of relapses during the therapy period. Additionally, the use of steroids and the total cumulative dosage were significantly reduced. Interestingly, in one study, the relapse rate was even higher after the discontinuation of the treatment [39], whereas Elmas et al. reported a reduced relapse risk 12 month after cessation of levamisole [40]. Basu and colleagues [39] retrospectively analyzed the data from children who had received either levamisole or MMF or tacrolimus. They verified that the relapse-free survival was higher after 30 months in patients treated with tacrolimus or MMF compared to levamisole (61.7% vs. 38.5% vs. 24%). Nevertheless, levamisole treatment resulted in a significant reduction in relapse rate compared to the relapse frequency in the year before treatment.

Table 1 and Table 2 summarized recent reports of levamisole for FRNS or SDNS in children and adults. As stated above, levamisole has been commonly used at 2.5 mg/kg/day every other day (maximum 150 mg). In all studies, levamisole was given at a dose of 35 mg/kg/month (2–2.5 mg/kg/day every other day), while a lower dosage seems to be less effective [31]. Some retrospective studies used levamisole every day. Many observational studies have been conducted, mainly to show the reduction of the recurrence rate in the treatment period after levamisole initiation compared with the time before levamisole. Although not many RCTs and meta-analyses had yet been performed, these studies also showed that levamisole reduced recurrence rate compared with placebo or steroid monotherapy. Some RCTs showed that levamisole was not inferior to MMF and cyclophosphamide. A meta-analysis was not sufficient to compare levamisole to MMF and cyclophosphamide (Table 3). We were not able to provide any data on accurate dose, timing, and duration of levamisole. The pharmacokinetic profile in children seems to be similar to the data available for adults, except for a documented slightly higher clearance rate in children [41]. A French trial (NEPHROVIR 3, NCT02818738) comparing the effects of levamisole with steroid at initial INS presentation for reducing relapse rate is currently ongoing.

## 4. Known Side Effects of Levamisole Treatment

Levamisole is a U.S. Food and Drug Administration pregnancy category C drug. It was withdrawn in the United States and Europe because of adverse effects (agranulocytosis and the risk of developing anti-neutrophil cytoplasmic antibody (ANCA)-positive vasculitis) and lack of clear indications [70]. However, the adverse effects of levamisole are mostly mild and transient, disappearing after its discontinuation. Common adverse effects include gastrointestinal symptoms (nausea, abdominal cramps), and pyrexia [71]. There were only a few serious adverse events documented in studies focusing on iNS. Single cases of gastrointestinal disturbances [31,41] or leukopenia [25,31] have been reported in the RCTs, probably caused by the use of levamisole. Most retrospective studies reported no side effects of levamisole [37,39,40], or only minor reversible effects such as rash, fever, abdominal pain, elevated liver enzymes, neutropenia or thrombocytopenia, which all disappeared after cessation of levamisole [36,52]. In their placebo-controlled multi-centric trial, Gruppen et al. showed that the use of levamisole was safe [28]. However, the definition of leukopenia was different in each study. In addition, recent studies have reported adverse events as neutropenia instead of leukopenia. Therefore, there is a clear limitation related to reporting all reported adverse effects together. Nevertheless, a comprehensive examination of reported studies indicates that the incidence of leukopenia varies from 0 to 20%. Of the 1391 cases that reported adverse effects of levamisole, 51 cases (leukopenia or neutropenia) had leukopenia (3.7%) (Table 4). This seems not much different from the incidence reported in other diseases [72]. Leukopenia related to the use of levamisole was spontaneously reversible after discontinuation of the treatment [28]. Levamisole can cause a variety of dermatologically adverse effects. It can cause lichenoid eruption, fixed dug eruptions, leg ulcers, purpura of the ears, and cutaneous necrosis [72]. Like other adverse events, symptoms resolved when levamisole treatment was stopped. In addition, leukoencephalopathy, skin necrosis, hyponatremia, acute coronary syndrome, pulmonary hypertension, granulomatosis with polyangiitis (GPA), and pyoderma gangrenosum were reported in single cases [73]. As levamisole is adulterated with cocaine, it may be used on the “black market” to enhance cocaine effects. The frequency of levamisole-induced GPA is increasing in the literature, with the proposed mechanisms having been recently highlighted [74].

Recent studies showed almost the same effect in adults with iNS [58]. Both KDIGO and Cochrane recommend levamisole with evidence grade 1b as an alternative medication to reduce the steroid load of patients with frequently relapsing iNS [69,77]. However, the precise pharmaceutical mechanism of this drug still needs to be evaluated. In addition, levamisole is used at a similar dose (2.0–2.5 mg/kg on alternate days, maximum 150 mg) to treat other diseases in adults [20,21,72], but few studies have investigated its efficacy in the management of nephrotic syndrome.

## 5. Mechanism Hypothesis

Proteinuria in nephrotic syndrome has been linked to various mechanistic dysfunctions in podocytes, leading to podocyte foot process effacement. MCNS is a pathological condition, which is characterized by subtle glomerular lesions causing massive and reversible proteinuria that is usually steroid sensitive. Glucocorticoids have historically been used as a first-line therapy in MCNS on the basis of their immunosuppressive function. However, it has been shown that podocytes serve as non-immunologic targets for several immunosuppressive drugs like cyclosporine and glucocorticoids [78]. Th2 cytokines (Interleukin (IL)-13, IL-4, IL-5) are increased in patients with nephrotic syndrome, and changes in vascular permeability have been reported, which are both involved in the complex pathogenesis of proteinuria in nephrotic syndrome [3,79,80,81]. Lai et al. showed that IL-13 transfected rats showed increased levels of proteinuria, raised serum cholesterol and decreased serum albumin levels and podocyte foot processes showed fusion mimicking MCNS [82]. The podocyte’s actin cytoskeleton is a discovered target for newer therapies. Calcineurin inhibitors, for example, block the dephosphorylation of the cytoskeleton component synaptopodin, and steroids increase actin polymerization and stability [83,84].

Levamisole is a synthetic imidazole derivative, which was used for its anthelmintic effects by acting as a nicotinic acetylcholine receptor agonist. Because various adverse effects including severe agranulocytosis in animals have been reported, levamisole is no longer used to treat any parasitic infections. Levamisole increases lymphocyte cyclic guanosine monophosphate (cGMP) by affecting cholinergic activity on T lymphocyte. Increased adenosine deaminase and free radicals act on both T and B lymphocytes. However, this drug has an additional immune-modulating action. It induces type 1 (Th1) and type 2 (Th2) immune responses through enhancing IL-18 activity [3]. It is also known that an enhanced Th1 or Th2 immune response is associated with several inflammatory and autoimmune diseases, such as SLE [85], asthma [86,87] and type 1 diabetes [88]. Therefore, levamisole is thought to have immunomodulatory potential. Szeto and colleagues showed that treatment with levamisole leads to a shift in the immune response in Brown Norway rats. This caused the augmentation of Th1 response and reduced the Th2 response [89]. Its effect is more pronounced on T lymphocyte than on B lymphocytes. Increased CD4^+^T/CD8^+^T Cell ratio and reduced IL-4 have been observed in levamisole-treated lichen planus patients and in a mouse model [87,90]. Experimental studies have shown that in mice with allergic rhinitis levamisole decreased IL-4, IL-5, and IL-13 mRNA and at the same time enhanced the expression of IL-12, IL-18 and interferon (IFN)-gamma mRNA [71]. IL-13 is an important Th2 cytokine involved in the induction of MCNS in a rat experiment model by the down-regulation of nephrin, podocin, and dystroglycan. It can also up-regulate B7-1 [65], and is altering the expression of zonula occludens-1 (ZO-1) in human podocytes [91]. In summary, the beneficial effect of levamisole on MCNS might be due to its suppressing effect on IL-13 expression. Upregulation of T helper-1 (Th1) cells and down-regulation of Th2 cells is considered to be the main mechanism for treating FRNS or SDNS patients [30,90,92]. Levamisole reduces immunoglobulin (IG) G, IG M, and circulating immune complex, which inhibit B cell activity [93]. Activated B cells caused podocyte foot process effacement via IL-4 [81]. In addition, total B cell increased in pediatric SSNS patients [94]. Although more mechanisms are still needed, not only T cells, but also B cells are considered to be therapeutic targets of levamisole.

Recent work has shown an additional direct effect of levamisole on podocytes. In an immortalized human podocyte culture, the authors were able to show that levamisole enhanced the expression and activity of the glucocorticoid receptor (GR). GR is present in glomerular cells including podocytes. Furthermore, levamisole protected podocytes in a puromycin aminonucleoside (PAN)-treated cell model. In this model, the effectiveness of levamisole is blocked by the GR antagonist mifepristone (RU486), suggesting that GR signaling is a critical target of levamisole’s action. Activated GR restoration of Bcl-2 and reduction of p53 in PAN treated podocytes resulting in inhibition of podocyte apoptosis [95]. Another study has shown that activated GR upregulates the expression of nephrin and Rho-A, leading to enhanced actin filament stability [96,97]. These data indicate that levamisole might be effective in nephrotic syndrome in adults as well as in children [58]. Those effects of levamisole are presented in Figure 1.

In summary, there is no universally accepted explanation how levamisole works in SSNS. Levamisole-induced changes in the immune-mediated response may cause reduction of relapses in predominantly immune mediated diseases. This may be one explanation for its ineffectiveness in cases of steroid-resistant nephrotic syndrome [98].

## 6. Summary and Conclusions

The effect of immunosuppressive agents in proteinuric glomerular diseases is attributed to immune-dependent mechanisms. However, recent studies have revealed that the anti-proteinuric effect of commonly used drugs might be related not only to their immunosuppressive actions, but also occur via a direct effect on podocytes. This also includes levamisole. Levamisole seems to be effective in reducing the risk of disease relapses in steroid sensitive forms of NS. In the reported studies, few serious side effects have been reported, compared to other steroid-sparing agents. There are hints that it is less effective compared to cyclophosphamide [34,69] and compared to MMF [39], but in comparison to both of these drugs, the adverse effects spectrum is beneficial. However, RCTs comparing different steroid-sparing agents head to head with levamisole in the treatment of SSNS are still lacking. Given the low frequency of side effects and that most adverse events improve with discontinuation of levamisole and KDIGO recommendations [77], levamisole could be a valuable alternative therapy for patients having frequent relapsing iNS. In addition, it has been shown that even after discontinuation of levamisole, long-term remission can be achieved. Therefore, levamisole should be considered a precious option in the treatment of SSNS, especially in cases of contraindications for immunosuppressive drugs. Regular monitoring of white blood count cells, renal function and testing of ANCA is advised in cases reporting symptoms of vasculitis.

## Figures and Tables

**Figure 1 jcm-08-00860-f001:**
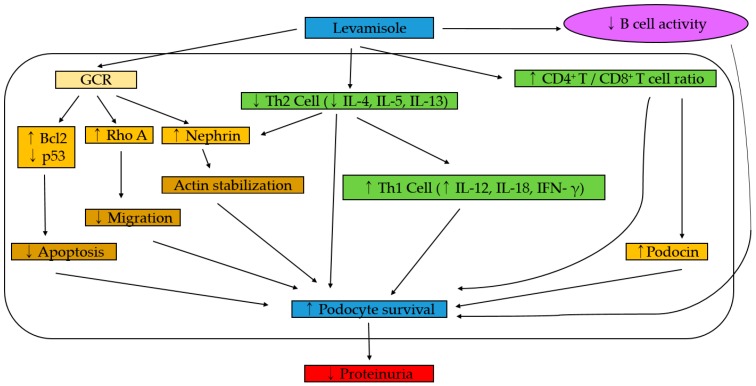
The summary effects of levamisole in iNS. Bcl2: B-cell lymphoma 2, CD: Cluster of differentiation, IFN-γ: interferon gamma, IL: interleukin, iNS: idiopathic nephrotic syndrome, GCR: glucocorticoid receptor, p53: tumor protein p53, Th Cell: The T helper cells.

**Table 1 jcm-08-00860-t001:** Observational studies of steroid-dependent or frequent relapsing nephrotic syndrome treated with levamisole.

Author, Year	Study Design	Comparison (Dose of Levamisole)	Total Patients	Relapse (%)	FU	Other Effect
Tanphaichitr P, 1980 [22]	Obs	(1.5–3.9 mg/kg twice per week)	7	-	1–6 mo	(33.5 ± 9.5 % vs. 69.3 ± 3.9%) ^a^
Niaudet P, 1984 [42]	Obs	(2.5 mg/kg twice per week)	30	-	9.9 mo	-
Mehta KP, 1986 [43]	Obs	(2.5 mg/kg qod or qd)	14	-	6–24 mo	6 Pts CR, 6 Pts PR
La Manna A, 1988 [44]	Obs	(2.5 mg/kg twice a week or qd)	13	-	7–29 mo	^b^
Srivastava RN, 1991 [45]	Obs	(5 mg/kg qod)	12	Before L (3.3/yr), After L (2.3/yr)	3 mo	-
Meregalli P, 1994 [46]	Obs	(5 mg/kg weekly)	10	Reduction SD 62–75%	-	-
Ksiazek J, 1995 [47]	Obs	-	22	18.2% relapse/36.4% not response	-	45.5% remission maintained
Ginevri F, 1996 [48]	Obs	(2.5 mg/kg qod)	20	45.5% remission maintained after L, 36.4% not respond to L	-	-
Bagga A, 1997 [23]	Obs	(2.5 mg/kg qod)	43	Before L (5.2/yr), After 2 years L (0.69/yr)	6–31 mo	-
Kemper MJ, 1998 [26]	Obs	(2.0 mg/kg qod)	25	Prior to L 0.5/month, During L 0.3/month	3–24 mo	-
Fu LS, 2000 [49]	Obs	(2–3 mg/kg qod)	27	Prior to L (5.74 ± 3.24/yr), During L (1.91 ± 2.0/yr)	6–24 mo	-
Alsaran K, 2001 [50]	Obs	L (2.5 mg/kg) vs. CPA	24	L (0.28/month), CPA (0.32/month)	-	^c^
Donia AF, 2002 [51]	Obs	(2.5 mg/kg qod)	20	25% in remission 12 mo, 75% relapses during Tx	12 mo	-
Al-Ibrahim AA, 2003 [52]	Obs	(2.5 mg/kg qod)	24	Prior to L (4/yr), During L (1.3/yr)	12–24 mo	-
Sumegi V, 2004 [53]	Obs	(2 mg/kg qd)	34	Prior to L (4.41/yr), During L (0.41/yr)	24 mo	Cumulative steroid dose 7564 mg/yr vs. 1472 mg/yr
Fu LS, 2004 [54]	Obs	(2–3 mg/kg qod or qd)	36	qod 2.01 ± 2.5/yr, qd 1.34 ± 2.1/yr	4–36 mo	-
Hafeez F, 2006 [55]	Obs	(2.5 mg/kg qod)	70	19 (27.14%) pts did not relapse on Tx	12 mo	^d^
Boyer O, 2008 [37]	Obs	(2.5 mg/kg 3 days/week)	10	Prior to L (6.0/yr), During L (0.0/yr)	48 mo	^e^
Madani A, 2010 [38]	Obs	(2.5 mg/kg qod)	304	Prior to L (2.0/yr), During L (1.1/yr)	1–22 yrs	Steroid dose was significantly reduced
Chen SY, 2010 [56]	Obs	(2–3.3 mg/kg qd)	15	1 pt CR, 10 pts No effect	3–20 mo	-
Elmas AT, 2013 [40]	Obs	(2.5 mg/kg 3 days/week)	29	Prior to L (4.0/yr), During L (0.0/yr)	12 mo	Proteinuria, annual SD reduced
Ekambaram S, 2014 [35]	Obs	(2 mg/kg qd)	97	Prior to L (2.4/yr), During L (1.3/yr)	6–24 mo	Steroid dose (4.1 g/m^2^ to 2.5 g/m^2^)
Skrzypczyk P, 2014 [57]	Obs	(2.5 mg/kg/qd or qod or twice a week)	21	-	-	^f^
Jiang L, 2015 [58]	Obs	(1.25 mg/kg qd)	15	Adult 14 pts reduction or stopping other medications	1–14 yrs	
Kuzma-Mroczkowska E, 2016 [36]	Obs	(2.5 mg/kg qod)	53	Prior to L (2.7/yr), During L (1.8/yr)	36 mo	8.8 mo to relapse on levamisole Tx
Alsaran, K, 2017 [59]	Obs	(2.5 mg/kg qod)	20	Prior to L (3.6/yr), During L (1.6/yr)	12 mo	Cumulative steroid dose was reduced
Abeyagunawardena, 2017 [60]	Obs	(2.5 mg/kg qod vs. qd)	58	L qod (163/yr), L qd (77/yr)	12 mo	M annual SD (mg/kg/yr) qod vs. qd (254 vs. 154)
Basu B, 2017 [39]	Obs	(2.5 mg/kg qod)	129	During L (1.7/yr) After L (2.8/yr)	30 mo	^g^

CPA: Cyclophosphamide, CR: complete response, FU: Follow up, L: Levamisole, M: mean, mo: months, Obs: observational study, PR: partial response, Pt: patient, qd: once daily, qod: every other day, SD: steroid dose, Tx: treatment, yr: year. ^a^ % of E-rosette before and after levamisole; ^b^ Response rate: levamisole 2.5 mg twice a week-4 of the 13 patients, levamisole 2.5 mg qd-2 patients of the 8 patients; ^c^ Levamisole (0.28) and cyclophosphamide (0.32) showed no significant difference in reduction of relapse rate; ^d^ Levamisole was more effective in older children (>5 years, vs. <5 years); ^e^ cumulative steroid 6067 mg/m^2^ to 2920 mg/m^2^; ^f^ Responder group was more frequently treated with steroid and levamisole and less frequently treated with cyclosporine A; ^g^ The relapse free survival was higher with tacrolimus than levamisole (61.7% vs. 24%).

**Table 2 jcm-08-00860-t002:** Randomized controlled trials of steroid-dependent or frequent relapsing nephrotic syndrome children treated with levamisole (RCT).

Author, Year	Inclusion	Exclusion	Comparison	Total Patients/Control	Relapse (%)	FU Weeks	Effect Size (RR, 95% CI)
BAPN, 1991 [25]	SSNS	NR	L (2.5 mg/kg qod) + steroid vs. Placebo + steroid (16 weeks)	61/30	L 57% (17/31) Placebo 87% (26/30)	16	0.63 (0.45–0.90)
Weiss, 1993 * [61]	FRNS, SDNS	Prior Tx with other IS within 6mo	L (2.5 mg/kg twice a week) + steroid vs. Placebo + steroid (6 mo)	48/26	L 95% (21/22) Placebo 81% (21/26)	24	1.18 (0.96–1.46)
Dayal, 1994 [62]	SSNS	SRNS	L (2–3 mg/kg twice a week) + steroid vs. steroid alone (48 weeks)	36/14	L 41% (9/22) Steroid alone 71% (10/14)	48	0.57 (0.31–1.05)
Rashid, 1996 * [63]	FRNS, SDNS	NR	L (2.5 mg/kg qod) + steroid vs. steroid alone (6 mo)	40/20	L 30% (6/20) Steroid alone 60% (12/20)	44	0.50 (0.23–1.07)
Sural, 2001 * [64]	FRNS, SDNS	NR	L (2.5 mg/kg qod) + steroid vs. steroid alone (6 mo)	58/28	L 27% (8/30) Steroid alone 82% (23/28)	48	0.32 (0.17–0.60)
Donia, 2005 [34]	SSNS	NR	L (2.5 mg/kg qod) + steroid vs. IV CPA + steroid	40/20	L 50% (10/20) CPA 55% (11/20)	96	0.91 (0.50–1.64)
Al–Saran, 2006 [65]	FRNS, SDNS	Prior Tx with IS	L (2.5 mg/kg qod) + steroid vs. steroid alone (1 yr)	56/24	Levamisole 9% (3/32) Steroid alone 50% (12/24)	48	0.19 (0.06–0.59)
Abeyagunawardena AS, 2006 * [66]	SSNS	NR	L (2.5 mg/kg qod) + steroid vs. No Tx	76/34	L 19% (8/42) Control 76% (26/34)	48	0.25 (0.13–0.48)
Gruppen, 2018 [28]	FRNS, SDNS	Unresponsiveness to cyclosporine or MMF	L (2.5 mg/kg qod) + steroid vs. Placebo (maximum 21 days)	99/49	L 66% (33/50) Placebo 86% (42/49)	48	0.30 (0.11–0.82)
Sinha, 2019 [30]	FRNS, SDNS	Prior Tx with other IS	L (2–2.5 mg/kg qod) vs. MMF	149/76	L 66% (48/73) MMF 59% (45/76)	48	0.79 (0.58–1.07)

BAPN: British Association for Pediatric Nephrology, CI: Confidence interval, CPA: Cyclophosphamide, FRNS: Frequently relapsing nephrotic syndrome, FU: Follow up, IS: immunosuppressive agents, L: Levamisole, MMF: Mycophenolate mofetil, mo: months, NR: not reported, qod: every other day, qd: every day, RCT: randomized controlled trial, RR: Risk ratio, SDNS: Steroid dependent nephrotic syndrome, SRNS: Steroid resistant nephrotic syndrome, SSNS: Steroid sensitive nephrotic syndrome, Tx: treatment, yr: year. * Data were extracted from the abstract.

**Table 3 jcm-08-00860-t003:** Summary of meta-analysis of RCTs on levamisole treatment in nephrotic syndrome.

Author, Year	Comparison	Outcome	No. of Studies	No. of Cases/Controls	Type of Metrics	Reported Summary Effect (95% CI)	Reported *p*-Value	Largest Effect (95% CI)	I^2^ (*p* Value)	No. of Significant Study/Total Study
Durkan AM, et al. 2001 [67]	L vs. placebo	Relapse (4–12 mo)	3	137/64	RR	0.60 (0.45–0.79)	0.0004	0.63 (0.45–0.90)	0.0% (0.84)	1/3
Durkan AM, et al. 2005 [68]	L vs. placebo/no treatment	Relapse (4–12 mo)	4	185/90	RR	0.71 (0.41–1.23)	0.23	0.63 (0.45–0.90)	86% (<0.001)	1/4
Hodson EM, et al. 2008. [69]	L vs. placebo	Relapse (4–12 mo)	6	317/148	RR	0.50 (0.25–0.99)	0.046	0.25 (0.13–0.48)	92% (0.046)	3/6
Hodson EM, et al. 2008 [69]	L vs. IV CPA	Relapse (E)	1	40/20	RR	0.91 (0.50–1.64)	NA	-	-	0/1
Pravitsitthikul N, et al. 2013 [31]	L vs. steroids or placebo or both, or no treatment	Relapse (4–12 mo)	7	375/176	RR	0.47 (0.24–0.89)	0.021	0.25 (0.13–0.48)	92% (<0.001)	4/7
Pravitsitthikul N, et al. 2013 [31]	L vs. IV CPA	Relapse (E)	2	97/47	RR	2.14 (0.22–20.95)	0.51	7.20 (0.96–53.89)	79% (0.03)	0/2

CPA: Cyclophosphamide, E: end of therapy, L: Levamisole, mo: months, NA: Significant, RCT: randomized controlled trial, RR: relative risk. Largest effect: Effect size (95% CI) and *p*-value from the largest study in each meta-analysis.

**Table 4 jcm-08-00860-t004:** Reported side effects of levamisole in nephrotic syndrome.

Author, Year	Levamisole Dose	Number	Leukopenia	GI Upset	Skin Rash	Arthritis	Other
BAPN, 1991 [25]	2.5 mg/kg qod	31	0	1	0	0	0
Weiss, 1993 * [61]	2.5 mg/kg twice a week	22	0	0	0	0	0
Sural, 2001 * [64]	2.5 mg/kg qod	30	1	0	0	0	0
Donia, 2005 [34]	2.5 mg/kg qod	20	0	0	0	0	14 ^a^
Al-Saran, 2006 [65]	2.5 mg/kg qod	32	0	1	0	0	0
Gruppen, 2018 [28]	2.5 mg/kg qod	50	8	1	0	1	26 ^b^
Sinha, 2019 [30]	2–2.5 mg/kg qod	73	0	18	2	0	0
Tanphaichitr P, 1980 [22]	1.5–3.9 mg/kg twice a week	7	0	0	0	0	0
Niaudet P, 1984 [42]	2.5 mg/kg twice a week	30	7	0	0	0	0
Metha KP, 1986 [43]	2.5 mg/kg qod	14	0	3	1	0	2 ^c^
La Manna A, 1988 [44]	2.5 mg/kg qd	13	0	1	1	0	1 ^d^
Prandota J, 1989 [75]	2.1–3.1 mg/kg qd	6	0	0	0	0	6 ^e^
Srivastava RN, 1991 [45]	5 mg/kg qod	12	1	0	1	0	0
Meregalli P, 1994 [46]	5 mg/kg weekly	10	1	0	1	0	0
Bagga A, 1997 [23]	2.5 mg/kg qod	43	0	0	0	0	0
Kemper MJ, 1998 [26]	2.0 mg/kg qod	25	2	1	1	0	0
Fu LS, 2000 [49]	2-3 mg/kg qod	27	7	0	0	0	0
Alsaran K, 2001 [50]	2.5 mg/kg qod	24	0	0	1	0	0
Donia AF, 2002 [51]	2.5 mg/kg qod	20	0	0	0	0	0
Al-Ibrahim AA, 2003 [52]	2.5 mg/kg qod	24	1	2	2	0	0
Sumegi V, 2004 [53]	2 mg /kg qd	34	5	0	0	0	0
Fu LS, 2004 [54]	2–3 mg/kg qd or qod	36	9	0	0	0	0
Hafeez F, 2006 [55]	2.5 mg/kg qod	70	0	0	0	0	0
Boyer O, 2008 [37]	2.5 mg/kg twice a week	10	0	0	0	0	0
Madani A, 2010 [38]	2.5 mg/kg qod	304	1	0	0	0	1 ^f^
Elmas AT, 2013 [40]	2.5 mg/kg twice a week	29	0	0	0	0	0
Ekambaram S, 2014 [35]	2 mg/kg qd	97	0	0	0	0	0
Jiang L, 2015 [58]	1.25 mg/kg qod	15	3	2	2	0	0
Kuzma-Mroczkowska,2016 [36]	2.5 mg/kg qod	53	1	3	9	0	6 ^g^
Alsaran K, 2017 [59]	2.5 mg/kg qod	20	4	0	0	0	0
Abeyagunawardena AS, 2017 [60]	2.5 mg/kg qd and qod	58	0	0	0	0	0
Basu B, 2017 [39]	2.5 mg/kg qod	129	0	0	0	0	3 ^h^
Youssef DM, 2018 [76]	2.5 mg/kg qod	23	0	0	0	0	0
Total		1391	51 (3.7%)	33 (2.4%)	21 (1.5%)	1 (0%)	59 (4.2%)

BAPN: British Association for Pediatric Nephrology, GI: gastrointestinal, qd: once daily, qod: every other day. ^a^ Respiratory infection (9), Scalp infection (2), Urinary tract infection (1), sialadenitis (1), personality change (1). ^b^ Cough (6), Nasopharyngitis (8), Pyrexia (10), ANCA-associated vasculitis (1), Reduced glomerular filtration rate (GFR) (1). ^c^ Transient hematuria (2). ^d^ Allergic conjunctivitis (1). ^e^ Thrombocytopenia (4), elevated liver transaminases (2). ^f^ Vertigo (1). ^g^ Elevated liver transaminases (3), arthralgia (2), thrombocytopenia (1). ^h^ Malaria (2), transient mood change (1). * Data were extracted from the abstract.

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
