# Peer review of "Levamisole in Children with Idiopathic Nephrotic Syndrome: Clinical Efficacy and Pathophysiological Aspects"

_jcm, 2019, doi:10.3390/jcm8060860_

Round 1
Reviewer 1 Report
Since the manuscript is titled" Levamisole in children with idiopathic nephrotic syndrome: clinical efficacy and pathophysiological aspects" it should be more focused on levamisole; introduction and effects of steroids, and other immunosupressives are discussed at too great lenghts; this loses focus of the reader interested in levamisole.
There are few spelling/grammar errors, for example "the golden standard" is usually referred to as gold standard.
"The aim of this review is to focus on the use of levamisole, which seems to be slightly less effective", seems slightly vague, and would suggest rephrasing it (could the numbers % be used for effectivness?).
In the statement "levamisole is adulterated with cocaine" the reason behind it should be explained as it rises concern that it may be used on the "black market" to enhance cocaine effects.
It would be helpful to include the Figure for visual help in delineating mechanism of action of levamisole.
Authors conclusion that "Given the low frequency of side effects, levamisole is a valuable alternative therapy for patients having frequent relapsing iNS." should be reconciled with "It was withdrawn in the United States and Europe because of adverse effects (agranulocytosis and the risk to develop anti-neutrophil cytoplasmic antibody (ANCA)-positive vasculitis) and lack of clear
188 indications [42]."
Along those lines authors should address timing of KDIGO (2013) recommendations to its use in the current practice.
It would also be helpful to the reader to know authors personal/center experience in treating iNS with levamisole; ..in our center..
Geographic and genetic differences in use/response to levamisol should be discussed in more details. Referring the reader to "The authors discussed different genetic backgrounds..." is a shortcoming. Is there any data in African-American and Hispanic populations?
Exclusion criteria in the studies on levamisol should be mentioned and discussed especially: unresponsiveness to cyclosporine or mycophenolate mofetil [28].
Author Response
Dear Reviewer
We would like to thank you and the reviewer of the JCM for taking the time to review our article (jcm-527242). We have made correction and clarification in the manuscript after going over the reviewer's comments. The changes are summarized below. We also attached the manuscript word file as "Revised manuscript with track changes".
Since the manuscript is titled" Levamisole in children with idiopathic nephrotic syndrome: clinical efficacy and pathophysiological aspects" it should be more focused on levamisole; introduction and effects of steroids, and other immunosupressives are discussed at too great lenghts; this loses focus of the reader interested in levamisole.
According to your comments. We shorten the introduction and effects of steroids and other immunosuppressives in “Revised Manuscript with track changes”.
Treatment with steroid is still the golden standard, (usually prednisone 60 mg/m2 for 4-6 weeks followed by 40 mg/m2 every second day for 4-6 weeks), and most forms (90%) of NS show remission within four weeks of treatment with steroids.
“However, severe side-effects (cushingoid changes, truncal obesity and buffalo humps), hyperglycemia, infection, delayed wound healing, abnormal bone metabolism, ophthalmic effects (including cataract formation, increase intraocular pressure, myopia, exophthalmos, papilledema, central chorioretinopathy, and subconjunctival hemorrhages), skin changes (skin atrophy, striae, ecchymoses, pustular acne, tinea incognito and Stevens-Johnson syndrome), peptic ulcer, adrenal suppression, myopathy, increase blood pressure, and psychiatric disturbances behavioral changes(including cognitive deficits, agitation, anxiety, distractibility, fear, hypomania, insomnia) are well recognized potentially serious adverse effects of corticosteroids [12-14].
Therefore, there is an increasing demand for alternative less harmful therapies. In the past, there has been extensive evidence for the use of so-called steroid sparing agents such as mycophenolic acid (MPA), rituximab, alkylating agents like cyclophosphamide and calcineurin inhibitors which are summarized and reviewed elsewhere [5, 15]. Cyclophosphamide has bladder toxicity (hemorrhagic cystitis), and other side effects such as alopecia, leukopenia, gonadal toxicity (infertility), bone marrow depression, carcinogenesis, and an increased risk of infection. Those adverse effects of cyclophosphamide are mostly more hazardous than corticosteroids [16]. Calcineurin inhibitors, cyclosporine and tacrolimus may lead to hypertension, neurotoxicity (tremor, seizure, akinetic mutism and so on), hirsutism, gingival hyperplasia, and nephrotoxicity. The nephrotoxicity is limiting its long-term use [17]. Mycophenolate mofetil (MMF) and MPA are newer immunosuppressive drugs that have fewer serious adverse effect but have the disadvantage of high therapy costs [18, 19].”
There are few spelling/grammar errors, for example "the golden standard" is usually referred to as gold standard.
->According to your comment we checked all spelling/grammar errors. Here is one of the changed sentences. “Treatment with steroid is still the golden standard, (usually prednisone 60 mg/m2 for 4-6 weeks followed by 40 mg/m2 every second day for 4-6 weeks), and most forms (90%) of NS show remission within four weeks of treatment with steroids.”
"The aim of this review is to focus on the use of levamisole, which seems to be slightly less effective", seems slightly vague, and would suggest rephrasing it (could the numbers % be used for effectivness?).
->As we shown in the table 1 and 2 most of levamisole studies were observation study and recent RCT study showed levamisole was not inferior to MMF (95% CI 0.79 (0.58-1.07)) and cyclophosphamide (95% CI 0.50-1.64) [Table 2]. Several meta analysis also showed no significant difference between levamisole and cyclophosphamide (2.14 (0.22-20.95)) [Table 3]. Therefore, we could not mention the number of percent in article. We changed to slightly “less effective” to “not inferior to” in in “Revised Manuscript with track changes”. “The aim of this review is to focus on the use of levamisole, which seems to be slightly less effective compared similar to other substances but exhibits lower number of adverse events and reactions.”
In the statement "levamisole is adulterated with cocaine" the reason behind it should be explained as it rises concern that it may be used on the "black market" to enhance cocaine effects.
->According to your comment, we changed “As levamisole is adulterated with cocaine because it may be used on the “black market” to enhance cocaine effects. The frequency of levamisole-induced GPA is increasing in literature with proposed mechanisms highlighted recently [48].” In “Revised Manuscript with track changes”.
It would be helpful to include the Figure for visual help in delineating mechanism of action of levamisole.
According to comments we inserted figure in “Revised manuscript with track changes”.
Authors conclusion that "Given the low frequency of side effects, levamisole is a valuable alternative therapy for patients having frequent relapsing iNS." should be reconciled with "It was withdrawn in the United States and Europe because of adverse effects (agranulocytosis and the risk to develop anti-neutrophil cytoplasmic antibody (ANCA)-positive vasculitis) and lack of clear 188 indications [42]."
->Side effects of levamisole are reversible (by discontinuing drugs) and can prevent via regular monitoring. Therefore, we changed “Given the low frequency of side effects and most adverse events improve with discontinuation of levamisole and KDIGO recommendations [50]. With regular monitoring neutrophil counts and ANCA level, levamisole is a valuable alternative therapy for patients having frequent relapsing iNS.” In “Revised Manuscript with track changes.”
Along those lines authors should address timing of KDIGO (2013) recommendations to its use in the current practice.
->According to your comments, we address the KDIGO Guideline (2012 KDIGO Clinical practice guideline for glomerulonephritis KI, 2012, 2(2), 139-274) recommended levamisole (evidence 1B) as corticosteroid-sparing agents, We changed “Given the low frequency of side effects and most adverse events improve with discontinuation of levamisole and KDIGO recommendations [50]. With regular monitoring neutrophil counts and ANCA level, levamisole is a valuable alternative therapy for patients having frequent relapsing iNS.” in “Revised Manuscript with track changes”.
It would also be helpful to the reader to know authors personal/center experience in treating iNS with levamisole; ..in our center..
->One of our co-author Markus J Kemper experienced some iNS patients with levamisole and those experiences are reported in some articles (Kemper MJ, Ammon O, Timmeermann K, Altrogge H, Muller-Wiefel DE: The treatment with levamisole of frequently recurring steroid-sensitive idiopathic nephrotic syndrome in children. Dtsch Med Wochenschr 1998, 123:239-342, [Table 1 and 4].
Geographic and genetic differences in use/response to levamisol should be discussed in more details. Referring the reader to "The authors discussed different genetic backgrounds..." is a shortcoming. Is there any data in African-American and Hispanic populations?
As shown Table 1, 4, observational study was performed from nearly all around the world (India, Iran, Saudi arabia, England, Turkey, Hungary, Italy, Netherland, Germany, Poland, Taipei, Tailand, South Korea, Canada, U.S.A., and so on). But most study did not showed detail of participants. Recently MP Gruppen, et al. reported that among 99 participants, 50 patients were Caucasian, and 2 patients were Black. Considering Canadian, USA and some Europe country reports, we could assume African-American, and Hispanic population may effect similar to other populations, but there were not enough evidences. In addition, currently ongoing French trial (NEPHROVIR 3, NCT02818738) aiming a testing the effect of levamisole associated with steroids at initial INS presentation for reducing relapse-rate at 12 months might be include lots of African-American and Hispanic population.
Exclusion criteria in the studies on levamisol should be mentioned and discussed especially: unresponsiveness to cyclosporine or mycophenolate mofetil [28].
As your comment we mentioned inclusion and exclusion criteria of studies on table 2. Most of study included FRNS, SDNS but exclusion criteria were different. Therefore, we added “Most of levamisole related study had different exclusion criteria. More studies are needed to confirm the effects of levamisole to unresponsiveness to other immunosuppressive agents.” in “Revised manuscript with track change”
Review 2
The authors performed an interesting and in depth review of the use of Levamisol in patients with Idiopathic neprotic syndrome. The review of Levamisol efficacy and tolerance is very complete and reflects well the current knowledge on this treatment. Considering the review of INS pathophysiology and how Levamisol could potentially act, the authors should better capture the controversial discussion around INS pathophysiology. Indeed, the authors affirm the T-cell origin of INS and insist on the involvement of various cytokines and on how Levamisol potentially mitigates cytokines production and effect. However, many recent data suggest the involvement of other types of immune cells (expecially B-cells). Although, the discussion of INS pathophysiology is beyond the scope of this review, the authors did report that Levamisol acts on both T and B cell and a more detailed discussion on its effect on B cell would be interesting.
We added “Levamisole reduce immunoglobulin (IG) G, IG M, and circulating immune complex that inhibit B cell activity [67]. Activated B cell cause podocyte foot process effacement via IL-4 [55]. In addition, total B cell were increased in pediatric SSNS patients [68]. Although more mechanisms are still needed, not only T cell but also B cells are involved in the levamisole treatment.” in “Revised Manuscript with track changes”.
Other specific comments:
- I would avoid specifying a steroid regimen since practices vary between countries. Moreover, most centers use a 4 s induction period (rather than 6 weeks) including the recently published PREDNOS trial from the UK.
According to your comments and Review 1’s comments, we changed “Treatment with steroid is still the golden standard, (usually prednisone 60 mg/m2 for 4-6 weeks followed by 40 mg/m2 every second day for 4-6 weeks), and most forms (90%) of NS show remission within four weeks of treatment with steroids. In the case of response towards steroids, iNS is classified as steroid sensitive nephrotic syndrome (SSNS), defined by the International Study of Kidney Disease in Children (ISKDC) [5].” in “Revised Manuscript with track changes.”
-I would not emphasized the potential direct effect of RTX on podocyte as this is very controversial and outside the scope of this review.
As your comment we also agree about the potential direct effect of RTX on podocyte. Therefore, We changed “However, it has been shown that podocytes serve as non-immunologic targets for several immunosuppressive drugs like cyclosporine ,RTX, and glucocorticoids” in “Revised Manuscript with track changes.”
- Amongst the potential use of Levamisol in INS, I would mention the currently ongoing French trial (NEPHROVIR 3, NCT02818738) aiming a testing the effect of Levamisol associated with steroids at initial INS presentation for reducing relapse-rate at 12 months.
According to your comment we rechecked the clinical trial and found one clinical trial about levamisole (French trial, NEPHROVIR 3, NCT02818378). We added ”Recently, French trial (NEPHROVIR 3, NCT02818738) to compare the effects of levamisole associated with steroid at initial INS presentation for reducing relapses rate is ongoing.” In “Revised Manuscript with track changes.”
Reviewer 2 Report
The authors performed an interesting and in depth review of the use of Levamisol in patients with Idiopathic neprotic syndrome. The review of Levamisol efficacy and tolerance is very complete and reflects well the current knowledge on this treatment. Considering the review of INS pathophysiology and how Levamisol could potentially act, the authors should better capture the controversial discussion around INS pathophysiology. Indeed, the authors affirm the T-cell origin of INS and insist on the involvement of various cytokines and on how Levamisol potentially mitigates cytokines production and effect. However, many recent data suggest the involvement of other types of immune cells (expecially B-cells). Although, the discussion of INS pathophysiology is beyond the scope of this review, the authors did report that Levamisol acts on both T and B cell and a more detailed discussion on its effect on B cell would be interesting.
Other specific comments:
- I would avoid specifying a steroid regimen since practices vary between countries. Moreover, most centers use a 4 s induction period (rather than 6 weeks) including the recently published PREDNOS trial from the UK.
-I would not emphasized the potential direct effect of RTX on podocyte as this is very controversial and outside the scope of this review.
- Amongst the potential use of Levamisol in INS, I would mention the currently ongoing French trial (NEPHROVIR 3, NCT02818738) aiming a testing the effect of Levamisol associated with steroids at initial INS presentation for reducing relapse-rate at 12 months.
Author Response
Dear Reviewer
We would like to thank you and the reviewer of the JCM for taking the time to review our article (jcm-527242). We have made correction and clarification in the manuscript after going over the reviewer's comments. The changes are summarized below.
The authors performed an interesting and in depth review of the use of Levamisol in patients with Idiopathic neprotic syndrome. The review of Levamisol efficacy and tolerance is very complete and reflects well the current knowledge on this treatment. Considering the review of INS pathophysiology and how Levamisol could potentially act, the authors should better capture the controversial discussion around INS pathophysiology. Indeed, the authors affirm the T-cell origin of INS and insist on the involvement of various cytokines and on how Levamisol potentially mitigates cytokines production and effect. However, many recent data suggest the involvement of other types of immune cells (expecially B-cells). Although, the discussion of INS pathophysiology is beyond the scope of this review, the authors did report that Levamisol acts on both T and B cell and a more detailed discussion on its effect on B cell would be interesting.
We added “Levamisole reduce immunoglobulin (IG) G, IG M, and circulating immune complex that inhibit B cell activity [67]. Activated B cell cause podocyte foot process effacement via IL-4 [55]. In addition, total B cell were increased in pediatric SSNS patients [68]. Although more mechanisms are still needed, not only T cell but also B cells are involved in the levamisole treatment.” in “Revised Manuscript with track changes”.
Other specific comments:
- I would avoid specifying a steroid regimen since practices vary between countries. Moreover, most centers use a 4 s induction period (rather than 6 weeks) including the recently published PREDNOS trial from the UK.
According to your comments and Review 1’s comments, we changed “Treatment with steroid is still the golden standard, (usually prednisone 60 mg/m2 for 4-6 weeks followed by 40 mg/m2 every second day for 4-6 weeks), and most forms (90%) of NS show remission within four weeks of treatment with steroids. In the case of response towards steroids, iNS is classified as steroid sensitive nephrotic syndrome (SSNS), defined by the International Study of Kidney Disease in Children (ISKDC) [5].” in “Revised Manuscript with track changes.”
-I would not emphasized the potential direct effect of RTX on podocyte as this is very controversial and outside the scope of this review.
As your comment we also agree about the potential direct effect of RTX on podocyte. Therefore, We changed “However, it has been shown that podocytes serve as non-immunologic targets for several immunosuppressive drugs like cyclosporine ,RTX, and glucocorticoids” in “Revised Manuscript with track changes.”
- Amongst the potential use of Levamisol in INS, I would mention the currently ongoing French trial (NEPHROVIR 3, NCT02818738) aiming a testing the effect of Levamisol associated with steroids at initial INS presentation for reducing relapse-rate at 12 months.
According to your comment we rechecked the clinical trial and found one clinical trial about levamisole (French trial, NEPHROVIR 3, NCT02818378). We added ”Recently, French trial (NEPHROVIR 3, NCT02818738) to compare the effects of levamisole associated with steroid at initial INS presentation for reducing relapses rate is ongoing.” In “Revised Manuscript with track changes.”